# Language Model Personalization via Reward Factorization

**Idan Shenfeld\*, Felix Faltings**\***& Pulkit Agrawal**
Massachusetts Institute of Technology
{idanshen,faltings,pulkitag}@mit.edu

**Aldo Pacchiano**
Boston University
pacchian@bu.edu

## Abstract

Modern large language models (LLMs) are optimized for human-aligned responses using Reinforcement Learning from Human Feedback (RLHF). However, existing RLHF approaches assume a universal preference model and fail to account for individual user preferences, limiting their effectiveness in personalized applications. We introduce a framework that extends RLHF to enable user personalization by leveraging the assumption that user preferences lie in a low-dimensional space. Instead of training a separate model per user, we represent user-specific rewards as a linear combination of base reward functions. Using only 10 user responses, our method can infer user-specific rewards and align LLM outputs accordingly. We validate our approach through experiments with both synthetic and real users, demonstrating significant personalization achieved by our method. In human evaluations, our method achieves a 67% win rate over default GPT-4o responses.

## 1 Introduction

A major driver of the success of modern large language models (LLMs) is their ability to generate responses aligned with human preferences. This alignment is typically achieved through Reinforcement Learning from Human Feedback (RLHF) (Ouyang et al., 2022), a technique that optimizes a reward function based on user preference data. Current approaches to RLHF assume a universal preference model across all users and cannot cater to individual user preferences, a key limitation to personalization (Casper et al., 2023; Sorensen et al., 2024).

User preferences vary widely across individuals and tasks. For example, one user might use an LLM as a professional assistant for work-related tasks, while another might use it as a virtual friend. Naively extending RLHF to cater to different user preferences, such as training a separate model for each user, is often infeasible. This is mainly due to the large amount of user-specific data required (typically thousands of data points (Gao et al., 2023)) and the significant computational cost of training and maintaining user-specific LLMs.

We present a framework called Personalization via Reward Factorization (PReF) that extends RLHF to support user personalization. PReF assumes that user preferences share structure (i.e., lie on a low-dimensional manifold (Rentfrow et al., 2011). Under this assumption, we represent the reward function of the $i^{th}$ user, $r_i$, which maps a prompt $x$ and response $y$ to a scalar, as a linear combination of $J$ "base" reward functions: $r_i = \sum_{j=1}^{J} \lambda_i^j \phi^j$. The coefficients $\lambda_i^j$ determine the contribution of each base function $\phi_j(x,y)$ to the $i^{th}$ user's reward function. We further assume that individual user preferences follow the standard Bradley-Terry formulation that computes rewards based on the user's pairwise rankings of responses (Bradley & Terry, 1952). Given a prompt $x$ and a pair of responses, $y_1, y_2$, the preference of $i^{th}$ user whether $y_1 \succ y_2$ is modeled as a Bernoulli random variable with parameter $\sigma\big(r_i(x, y_1) - r_i(x, y_2)\big)$. With these assumptions, PReF reduces the problem of personalization to estimating user-specific coefficients $\lambda_j^i$, which is simpler and more data-

---

*equal contribution

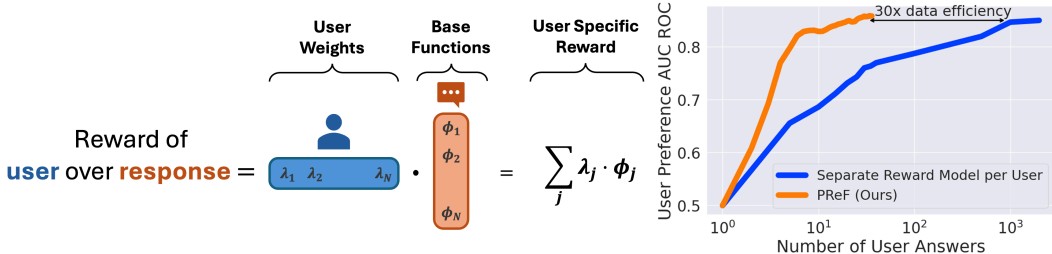

Figure 1: We factorize each user's personal reward as a linear combination of base functions. The linear structure enables us to perform personalization in an efficient manner, needing up to x30 fewer answers from the user to achieve the same performance as the standard RLHF approach.

efficient than learning separate reward models per user. PReF, therefore, greatly reduces the need for user-specific data and computation.

Previous work on LLM alignment developed methods to combine a set of pre-defined reward functions linearly but did not focus on personalization (Han et al., 2024; Guo et al., 2024; Yang et al., 2024b). In particular, these approaches do not address the core problems necessary for data and compute efficient user personalization: (1) determining the base reward functions and (2) identifying the user-specific combination of base reward functions that personalize responses. Our work addresses these questions.

In PReF, we first collect preference data in which each pair of responses to the same prompt is annotated with the user's preference (i.e., which response is preferred) and the user's identity. We use this dataset to learn the base reward functions. Once the base reward functions are determined, the next step is to infer the coefficients for each new user. To achieve this, we generate a sequence of questions and a pair of responses and ask the user to indicate which response they prefer. Based on the responses, we estimate the user coefficients and, thus, its specific reward function.

The challenge in estimating the user's coefficients is to do so with a minimum number of questions presented to the user. To do so, we adopt an active learning approach where the sequence of answers is adaptive to the user, meaning that the questions are selected based on the user's prior responses to efficiently refine their preference model. Specifically, we select a question and responses that minimize the uncertainty of the user's coefficients. We adapt and extend results from the logistic bandit literature to efficiently compute uncertainty scores of response pairs. Using our method, we can determine the user coefficients using only 10-20 questions.

Once the user-specific reward function has been identified, the next step is to align the LLM to it. We leverage recent advances in inference-time alignment methods (Han et al., 2024; Yang et al., 2024b; Rame et al., 2024) that can generate reward-aligned responses from an LLM at deployment without modifying the weights of the LLM. This allows for efficient, scalable adaptation to individual users without requiring costly model updates.

We validate our framework with extensive experiments. On synthetic data where we use LLMs to mimic real users, our framework outperforms standard RLHF by a large margin. Our personalized reward model needs only five samples from new users to perform better than a standard reward model. Finally, we demonstrate our method's ability to cater to real human preferences. In our experiment, we aligned GPT4o with real users, achieving 67% win rate over the model default answers.

## 2   Related Work

Personalization of LLMs has become an important research direction, enabling models to better serve individual users' needs (Sorensen et al., 2024; Kirk et al., 2024b; Zhang et al., 2024). Broadly, personalization can take several forms: incorporating user-specific knowledge, fine-tuning models to develop domain expertise, or adjusting response styles to align with user preferences (Ning et al., 2024; Wu et al., 2024; Richardson et al., 2023; Kirk

et al., 2024a; King & Cook, 2020). Our work focuses on the last category—personalization through user-specific preference alignment.

A leading approach for aligning LLMs with human preferences is Reinforcement Learning from Human Feedback (RLHF), first introduced by (Christiano et al., 2017) and further refined in later works (Ouyang et al., 2022; Ziegler et al., 2019; Stiennon et al., 2020; Bai et al., 2022b). RLHF trains a reward model using datasets of response pairs annotated with human preferences (Wang et al., 2024a), often requiring thousands to hundreds of thousands of labeled examples (Gao et al., 2023).

To improve the alignment process, researchers have proposed decomposing human preferences into distinct aspects, such as helpfulness, harmlessness, and factuality (Bai et al., 2022a; Wang et al., 2024b; Dorka, 2024). In these approaches, a separate reward function is trained for each of these properties and reinforcement learning is performed on their weighted sum. This decomposition facilitates learning each how to maximize each property independently and allows for control over their balance in downstream applications. Extending this idea, multi-reward formulations have been proposed for personalization, where each user has a different combination of these reward functions (Guo et al., 2024; Zhou et al., 2023; Yang et al., 2024b; Wang et al., 2024c). Although this supports personalization, a key limitation is that it typically requires training separate models for each reward combination.

Several approaches have tackled this challenge by reweighting reward functions at inference time, allowing for dynamic model adaptation without retraining (Han et al., 2024; Chen et al., 2024b; Khanov et al., 2024; Mudgal et al., 2023). Others have trained separate models for different reward functions and later combined them in weight space (Jang et al., 2023; Rame et al., 2024). However, these methods rely on the assumption that reward functions are pre-defined and that user preferences are explicitly specified. In contrast, our work develops personalization algorithms that relax these constraints, enabling more flexible and adaptive model behavior.

## 3 Preliminaries

Our objective is to generate responses $y$ to a given prompt $x$ that align with the preferences of an individual user. To capture these preferences, we assume that each user $i$ has a reward function $r_i(x, y)$, which quantifies how well a response $y$ satisfies the user's expectations for a given prompt $x$. In order to produce user-aligned responses to a given prompt, it is imperative to learn the user-specific reward function and then use the LLM to generate responses that maximize it.

Instead of requiring users to assign explicit scores to responses, we follow common practice (Ouyang et al., 2022) and rely on pairwise comparisons, where users indicate their preference over a pair of responses. We adopt the Bradley-Terry (BT) choice model (Bradley & Terry, 1952; Christiano et al., 2017) for pairwise comparisons. The BT model defines the probability that user $i$ prefers response $y^1$ over $y^2$ as:

$$p(y^1 \succ y^2 | x, i) = \sigma(r_i(x, y^1) - r_i(x, y^2)) \tag{1}$$

where $p(y_1 \succ y_2 | x, i)$ is the probability that user $i$ prefers $y_1$ over $y_2$, and $\sigma(w) = \frac{1}{1+e^{-w}}$ is the sigmoid function.

In standard RLHF, a single, global reward function $r(x, y)$ is learned by maximizing the likelihood of all pairwise comparisons across the dataset. This is formalized as optimizing the objective:

$$\mathcal{L}(\theta) = \sum_{(y^1, y^2, x)} \log p(y^1 \succ y^2 | x; \theta) \tag{2}$$

where $\theta$ represents the reward model. This objective assumes homogeneous preferences across users, treating all pairwise comparisons as arising from the same reward function $r(x, y)$.

While effective for general alignment tasks, this approach fails to account for user-specific variations in preferences. By aggregating data from all users into a single global reward model, standard RLHF overlooks individual differences, potentially leading to suboptimal personalization.

## 4 The PReF framework

In this work, we model the reward function of an individual user $i$ as a linear combination of $J$ base reward functions $\phi(x, y) = [\phi^1(x, y), \phi^2(x, y), \ldots, \phi^J(x, y)]^\top \in \mathbb{R}^J$, where each $\phi^j(x, y) \in \mathbb{R}$ quantifies the score of the response $y$ to the prompt $x$ based on the $j$-th base function.

Each user $i$ is characterized by a preference vector $\lambda_i = [\lambda_i^1, \lambda_i^2, \ldots, \lambda_i^J]^\top \in \mathbb{R}^J$, where $\lambda_i^j$ represents the weight that user $i$ assigns to the $j$-th base reward function. The overall reward for user $i$ is then defined as:

$$r_i(x, y) = \sum_{j=1}^{J} \lambda_i^j \cdot \phi^j(x, y) = \lambda_i^\top \phi(x, y) \tag{3}$$

This formulation provides a compact representation of user-specific preferences, with the weights $\lambda_i$ capturing the unique importance each user assigns to the $J$ base reward functions. Plugging it into Equation 1 gives us the PReF pairwise preference model [1]:

$$p(y^1 \succ y^2 | x, i) = \sigma(\lambda_i^\top \phi(x, y^1) - \lambda_i^\top \phi(x, y^2)) \tag{4}$$

In practice, we train a neural network to estimate $\phi$. This network gets a concatenation of the prompt $x$ and response $y$ as input and outputs a $J$-dimensional vector. To train this neural network, we assume access to a pairwise preference dataset where each prompt is annotated by multiple users, each with individual preferences. Formally, the dataset is represented as $\{x_n, y_n^1, y_n^2, i_n, A_n\}_{n=1}^N$, where $i_n$ is the index of the user providing the annotation, and $A_n \in \{0, 1\}$ denotes the user's binary preference, with $A_n = 1$ indicating that the user prefers $y_n^1$ over $y_n^2$. Given $U$ different users and $M$ pairs of responses, we can represent the dataset in a matrix form:

$$A \sim \text{Bernoulli}(P), \quad P = \sigma(\Lambda^\top \Phi)$$

where $A \in \mathbb{R}^{U \times M}$ contains the observable binary preferences in matrix form, $P \in \mathbb{R}^{U \times M}$ contains the preference probabilities as per Equation 4, $\Lambda \in \mathbb{R}^{J \times U}$ is the matrix of user preference vectors, and $\Phi \in \mathbb{R}^{J \times M}$ is the matrix of base reward function embeddings for all response pairs.

Such a representation of reward function enables us to leverage existing algorithms that can adapt the response of the large language models to a linear combination of multiple reward terms at deployment time (Han et al., 2024; Chen et al., 2024b; Khanov et al., 2024; Mudgal et al., 2023).

### 4.1 Learning the Base Functions

We train the base reward function model $\phi$ and user embeddings $\lambda$ using the Maximum Likelihood Estimator (MLE) objective of Equation 4:

$$\begin{aligned}
\mathcal{L}(\lambda, \phi) = \sum_{n=1}^{N} & A_n \cdot \log \sigma(\lambda_{i_n}^\top \phi(x_n, y_n^1, y_n^2)) \\
& + (1 - A_n) \cdot \log(1 - \sigma(\lambda_{i_n}^\top \phi(x_n, y_n^1, y_n^2))),
\end{aligned} \tag{5}$$

Unlike standard MLE in RLHF (Eq. 2), our formulation introduces significant challenges. First, the number of users parameters $\lambda$ scales with the number of users in the training

---

[1]For simplicity of notation, when dealing with pairwise comparisons of responses $y^1$ and $y^2$ for the same prompt $x$, we will denote them as $\phi(x, y^1) - \phi(x, y^2) = \phi(x, y^1, y^2)$.

set, increasing complexity. More critically, the reward model exhibits bilinear dependency between $\lambda_i$ and $\phi(x, y^1, y^2)$, which makes the optimization landscape non-convex with many local minima. This coupling makes the optimization sensitive to initialization and prone to degenerate solutions (e.g., trivial or uninformative user vectors). Results in Section 5 show that such instability leads to high variance in the performance of the trained model.

To mitigate these instabilities, we leverage the linear structure of our framework. Specifically, we recognize that Since $\sigma^{-1}(P) = \Lambda^\top \Phi$, when the preference probability matrix $P$ is known, we can recover $\Lambda^\top \Phi$ by applying the inverse sigmoid function and reducing the problem of learning ($\Lambda$) and ($\Phi$) to a matrix factorization problem. However, since $P$ is unknown and and only sparse binary observations in $A$ available, the learning task becomes an instance of Logistic Matrix Factorization (Johnson et al., 2014) problem.

Using these insights, we propose a two-step approach to overcome the instability challenges when training $\phi$ (see formal description in Algorithm 1):

1. **Initialization via SVD:** We initialize training using Singular Value Decomposition (SVD) of the observed annotation matrix $A$, treating it as a noisy proxy for the underlying preference probability matrix $P$. The low rank outputs of the SVD are used as initialization for $\Lambda$ and $\Phi$, offering a structured initialization that reduces sensitivity to random starting conditions. While the binary nature of $A$ introduces noise, SVD still captures the dominant components of $P$, providing a meaningful starting point.

2. **Refinement via MLE:** Although SVD provides a strong initialization, it does not directly optimize the likelihood of observed preferences. Therefore, we refine the factorization using the MLE objective. In our experiments we have found that the magnitude of either $\phi$ or $\lambda$ tends to be big, which hurts downstream performance. We tracked the core of the problem to the fact that the reward factorization $\Lambda^\top \Phi$ is not unique. For any invertible matrix $R$, we have $\Lambda^\top \Phi = \Lambda^\top R^{-1} R \Phi$. Therefore, to stabilize the training we add L2 regularization of the user vectors $\lambda$ to the MLE objective. This prevents extreme parameter values, reduces instability, and addresses scale ambiguity in matrix factorization. As a result, training converges more consistently.

This combination of SVD initialization and regularized optimization addresses the instability issues associated with bilinear optimization and ensures a consistent and stable learning process.

## 4.2 Adaptation to a New User

After learning the base reward functions, the next step is to estimate the weight vector $\lambda$ for a new user based on their preferences. The challenge is to do this efficiently, requiring as little user feedback as possible to reduce the effort required from the user. This process involves iteratively collecting pairwise feedback and refining the estimate of $\lambda$.

In each round $t \in \{1, ..., T\}$, we sample a prompt $x_t$ and use an uncertainty-based selection strategy to determine a pair $y_t^1, y_t^2$ of responses to provide the user. We aggregate the prompt, responses, and the user preference $A_t$ into a dataset and use it to estimate the user preference using the regularized MLE objective:

$$
\begin{aligned}
\mathcal{L}(\lambda) = \sum_{s=1}^{t} & A_s \cdot \log \sigma(\lambda^\top \phi(x_s, y_s^1, y_s^2)) \\
& + (1 - A_s) \cdot \log(1 - \sigma(\lambda^\top \phi(x_s, y_s^1, y_s^2))) + \frac{\beta}{2} \|\lambda\|_2^2
\end{aligned}
\tag{6}
$$

Where $\beta$ is a hyperparameter that controls the weight of the L2 regularization. Given that during adaptation the features $\phi$ are known, the problem of inferring $\lambda$ is a plain logistic regression problem which is concave (Kleinbaum et al., 2002) that does not suffer the instabilities that we had while learning the features.

Our strategy to improve data efficiency is to choose the next response pair that maximizes uncertainty, a standard approach in active learning (Ren et al., 2021). In this work the

uncertainty for a candidate prompt-response pair $(x, y_1, y_2)$ is defined as the largest potential prediction error:

$$U_t(x, y^1, y^2) = \max_{\lambda \in \mathcal{C}} |\lambda^T \phi(x, y^1, y^2) - \lambda_t^T \phi(x, y^1, y^2)| \tag{7}$$

where $\lambda_t$ is the MLE estimate of $\lambda$ at round $t$, and $\mathcal{C}$ is a confidence set for $\lambda^*$ (the true user preferences). Intuitively, this metric quantifies how much the predicted preference for the response pair could vary given uncertainty in $\lambda$.

For logistic regression, the tightest known confidence set (Faury et al., 2020) can be expressed using the Hessian matrix of the log-likelihood function, $H_t(\lambda)$:

$$H_t(\lambda) = \sum_{s=1}^{t-1} \sigma'(\lambda^\top \phi(x_s, y_s^1, y_s^2))\phi(x_s, y_s^1, y_s^2)\phi(x_s, y_s^1, y_s^2)^\top + \beta I \tag{8}$$

where $\sigma'$ is the derivative of the sigmoid function. Using this Hessian, we define the confidence set:

**Lemma 4.1.** *(Faury et al. (2020), Lemma 11) Let $\mathcal{E}_t(\delta) = \{\lambda \in \mathbb{R}^d \mid \|\lambda - \lambda_t\|_{H_t(\lambda)} \le \gamma_t(\delta)\}$ where $\gamma_t(\delta) = \mathcal{O}\left(d \log\left(\frac{t}{\delta}\right)\right)$, and assume $\|\phi\| \le 1$. The following holds with probability at least $1 - \delta$ for all $t \in \mathbb{N}$.*

$$\lambda^* \in \mathcal{E}_t(\delta).$$

While $\mathcal{E}_t(\delta)$ is theoretically tight, it is computationally infeasible to directly solve Equation 7 under this constraint since we do not have a way to avoid iterating over every $\lambda \in \mathcal{E}_t(\delta)$. To address this, we introduce a relaxed confidence set $\mathcal{E}_t^{exp}(\delta)$ that provide a simple solution to Equation 7. The new confidence set is constructed by replacing the Hessian $H_t(\lambda)$ with the Hessian evaluated at $\lambda_t$:

**Lemma 4.2.** *Let $\mathcal{E}_t^{exp}(\delta) = \{\lambda \in \mathbb{R}^d \mid \|\lambda - \lambda_t\|_{H_t(\lambda_t)} \le \zeta_t(\delta)\}$ where $\zeta_t(\delta) = \mathcal{O}(e^d d \log(\frac{t}{\delta}))$ The following holds with probability at least $1 - \delta$ for all $t \in \mathbb{N}$.*

$$\lambda^* \in \mathcal{E}_t^{exp}(\delta).$$

Using the expanded confidence set[2], the uncertainty metric simplifies to:

**Lemma 4.3.** *The following holds with probability at least $1 - \delta$ for all $t \in \mathbb{N}$:*

$$U_t(x, y^1, y^2) = \left\|\phi(y^1, y^2, x)\right\|_{H_t^{-1}(\lambda_t)} \cdot \zeta_t(\delta).$$

Therefore, to ensure that we choose $y_1, y_2$ that we are most uncertain about, we solve the following:

$$\max_{y^1, y^2} \left\|\phi(x, y^1, y^2)\right\|_{H_t^{-1}(\lambda_t)} \tag{9}$$

The solution for $\phi$ is the eigenvector of $H_t^{-1}(\lambda_t)$ corresponding to its largest eigenvalue (Hamming, 2012), which we will denote $\nu$. To obtain a response pair $y^1, y^2$ such that $\phi(x, y^1, y^2) = \nu$ we will use an inference time alignment algorithm to generate a response $y^1$ such that $\phi(x, y^1) = \frac{1}{2}\nu$ and $\phi(x, y^2) = -\frac{1}{2}\nu$. See full description of the procedure in Algorithm 2.

---

[2]While the expanded confidence set introduces an exponential dependence on the dimension, our response selection strategy (Equation 9) is not explicitly affected by this. Empirically, we observe that the approach performs well in practice, suggesting that more refined analytical techniques could potentially yield a tighter bound.

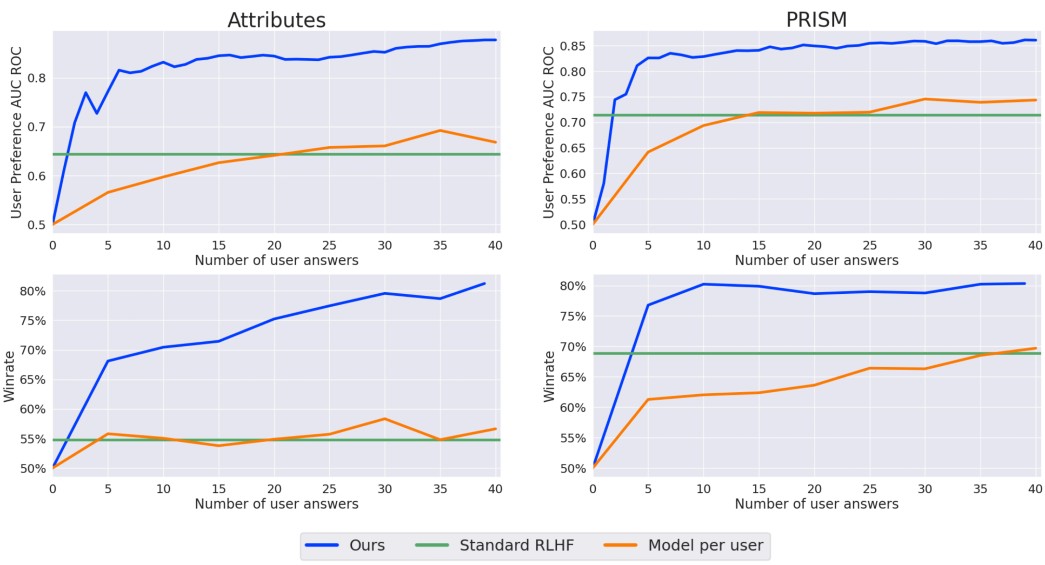

Figure 2: ROC AUC and winrates for varying number of user answers on the Attributes (left) and PRISM (right) datasets. Our method quickly achieves high ROC AUC and winrates, outperforming baselines by a large margin.

# 5 Experiments

**Datasets.** We test our method using the following datasets (more details in Appendix B):

- **Attributes**. To test personalization, we introduce a dataset that simulates diverse user preferences using LLMs as a roleplay judge (Dong et al., 2024; Zheng et al., 2023). We defined seven preference attributes, each with a positive and negative trait. For example, the attribute *length* corresponds to users who either prefer verbose or concise responses. Each user is assigned two randomly sampled traits, resulting in 84 unique users. Preference data for each user is collected over responses generated using prompts from the AlpacaEval dataset (Li et al., 2023), resulting in 100 user preferences per prompt.
- **PRISM**. We use the PRISM dataset (Kirk et al., 2024b), which contains user preferences for LLM-generated content from 1.5K global respondents across 3K prompts, often showing significant disagreement. Since the original dataset lacks overlap between users and prompts—required for our method—we augment it with synthetic annotations following the PERSONA protocol (Castricato et al.). PERSONA uses LLMs as judges, shown to correlate well with human preferences. This augmentation resulted in 50 user preferences per prompt.

**Training and Evaluation Protocol.** All experiments use Qwen 2.5, an open-source state-of-the-art model family (Yang et al., 2024a). Unless stated otherwise, we use the 0.5B variant as the reward model backbone with a single-layer linear head. Each experiment is repeated 10 times with different random seeds, and we report aggregated results. To demonstrate compatibility with various alignment methods, we use ChatGPT-4 with multi-objective Best-of-N for the *Attributes* dataset and Qwen2.5 7B with VAS (Han et al., 2024) for *PRISM*. Hyperparameters and further training details appear in Appendix C.

Each dataset is split into four parts: a train set, a validation set (same users, new prompts), a calibration set (new users, same prompts), and a test set (new users and prompts). We first train the base reward functions on the train set. To evaluate PReF 's personalization ability, we infer test users' preference coefficients using the calibration set. Final evaluation is on the test set. We use two metrics: (1) **Winrate**, measuring how often responses aligned with the learned reward function outperform non-personalized responses (evaluated via LLM-as-a-Judge); and (2) **User Preference AUC-ROC**, measuring how well the reward function predicts user choices from response pairs, independent of downstream alignment.

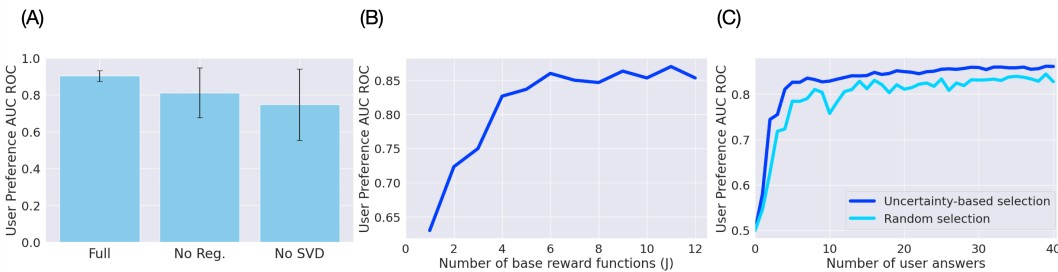

Figure 3: (A) Effect of L2 regularization and SVD initialization on model performance. We see that both choices are crucial to reduce instabilities in training. (B) Increasing the feature dimension $J$ leads to better performance. (C) PReF's uncertainty-based selection of response pairs to obtain user preferences outperforms the naive strategy of random selection.

## 5.1 The Benefits of personalization

To assess PReF 's ability to capture personalized preferences, we compare it to two baselines: *Standard RLHF*, which trains a single reward model assuming homogeneous preferences, and *Model per User*, which trains a separate reward model for each user.

The results in Figure 2 shows that across both datasets, PReF (blue) clearly outperforms Classic RLHF (green). With fewer than 10 user responses, PReF achieves a 10–15% AUC-ROC gain, showing strong personalization from limited data. A similar boost is seen in win rate: about +10% on PRISM and +25% on Attributes. In addition, the Model per User baseline (orange) performs poorly due to the inability to leverage shared structure across users, confirming that training separate models per user is impractical in real-world settings. Figure 8 in the Appendix shows the performance of the Model per User baseline for a much larger number of user answers. It shows that our approach requires 25x less data to achieve the same performance.

## 5.2 Can PReF capture the preferences of real humans?

We validated our framework on real users by conducting a human evaluation study focused on adapting to new users with a pre-trained set of features.

We use the base functions learned from the synthetically generated *Attributes* dataset. We recruited 28 volunteers, each shown 30 prompts from the test set with two generated answers, and asked to choose their preferred response. The first 15 comparisons were used to learn the user's preferences and the remaining 15 for evaluation. The user was not aware of this distinction. In evaluation, one response was always generated as a baseline using GPT-4o, while the other was a personalized version using the learned user preference. We computed the winrate of the personalized answers over the baseline answers. Additional details are given in Appendix D.

We found that our method achieved a **67% winrate**, with a 95% confidence interval of $[57.4\%, 76.6\%]$ winrate. That shows that by tailoring the responses to each user's preferences PReF improves over GPT-4o. This improvement is notable given that GPT-4o has already been aligned to general human preferences, and given that we used very simple features derived from our synthetic data. Moreover, the user preferences were learned from just 15 interactions with the user.

## 5.3 How PReF performs against other personalization frameworks?

We benchmark PReF against other LLM personalization frameworks - Variational Preference Learning (VPL) (Poddar et al., 2024) and Pluralistic Alignment (PAL) (Chen et al., 2024a). VPL conditions the reward model on encoding of the user responses and training it to achieve in-context learning ability. PAL represents each user as a vector in a latent space and defines the reward of a response as its distance from this point.

To assess performance, we evaluate these methods on the Attributes dataset, measuring AUC-ROC for unseen responses after collecting 5, 10, or 20 answers from a new user. For a fair comparison, we ensure that each method undergoes the same number of hyperparameter tuning experiments, with results averaged over five random seeds. The results, presented in Table 1, show that while VPL performs well, PReF outperforms it at 10 and 20 user responses. This shows that in-context learning has a hard time utilizing a large number of examples.

|  | Number of User's Responses | | |
|---|---|---|---|
|  | 5 | 10 | 20 |
| PReF (Ours) | 0.77 | **0.83** | **0.85** |
| VPL (Poddar et al., 2024) | **0.78** | 0.80 | 0.80 |
| PAL (Chen et al., 2024a) | 0.56 | 0.59 | 0.61 |

Table 1: AUC-ROC over responses from the initial model. Our method outperforms other proposed frameworks for efficient personalization of LLMs.

Beyond reward-learning approaches, we also compare against two widely used technique for personalizing LLM outputs through prompts. The first is the user-provided system prompt, where we use the prompts in the PRISM dataset (Kirk et al., 2024b) that were written by each participant. The second is in-context learning baseline, concatenating the user-specific dataset (prompt, responses, and preferences) into one prompt. We measure the win rate of responses generated from PReF against these generated from the baselines. PReF achieves a win rate of 71.9% against the system prompts baseline, demonstrating a clear advantage in capturing individual user preferences. For the in-context learning baseline, PReF achieves 56.1% win rate at 5 responses, 62.7% at 10 responses, and 68.4% at 20 responses.

## 5.4 Ablations

Our optimization framework introduces bilinear dependencies between learning the base reward functions and the user coefficients, that can lead to instability and sensitivity to initialization. To address this, we incorporate SVD-based initialization to provide a structured starting point and L2 regularization to stabilize the MLE optimization (Section 4.1).

Figure 3 (A) validates the importance of these components by comparing our full method (*Full*) to two ablations: (1) *No Reg.*, which removes L2 regularization, and (2) *No SVD*, which replaces SVD-based initialization with random embeddings. The figure reports the mean and standard deviation of the mean over 10 models trained on the same data with different seeds. Removing SVD leads to significantly higher variance, particularly in the new user setting, highlighting its role in reducing sensitivity to random initialization. Similarly, without L2 regularization of the user's coefficients, the standard deviation of the mean also increases, suggesting that regularization prevents overfitting and stabilizes optimization.

Additionally, we evaluate the benefits of our active learning approach in determining user weights. In Figure 3 (C), we compare our method to a baseline where questions presented to the user are chosen at random. The results clearly demonstrate the advantage of our approach: our method achieves x2.7 increase in efficiency - getting the same performance with just 15 samples that random selection requires over 40 samples to reach.

## 6 Discussion

In this work we propose a method to quickly adapt LLM generations to personal user preferences. We model each user's preferences as a linear combination of base features. Given fixed features, we can learn a new user's weights from a small number of preference feedbacks using a logistic bandit algorithm. These weights can then be used to adapt LLM generations without the need for retraining. Moreover, we can effectively learn the base

features from data, circumventing the need to predefine them. A major limitation of our work is that most available datasets do not contain preference feedback from multiple users for the same pair of generations, making the learning problem ill-posed. We thus had to limit many of our experiments to synthetically generated data using LLMs to roleplay human users. Collecting a large dataset with such overlapping annotations is an important goal for future work.

## Acknowledgements

We thank members of the Improbable AI Lab for helpful discussions and feedback. We are grateful to MIT Supercloud and the Lincoln Laboratory Supercomputing Center for providing HPC resources. This research was supported in part by Hyundai Motor Company, Qualcomm Innovation Fellowship, an AWS MLRA research grant, ARO MURI under Grant Number W911NF-23-1-0277, DARPA Machine Common Sense Program, ARO MURI under Grant Number W911NF-21-1-0328, and ONR MURI under Grant Number N00014-22-1-2740. The views and conclusions contained in this document are those of the authors and should not be interpreted as representing the official policies, either expressed or implied, of the Army Research Office or the United States Air Force or the U.S. Government. The U.S. Government is authorized to reproduce and distribute reprints for Government purposes, notwithstanding any copyright notation herein.

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

# A    Proofs and Derivations

**Lemma 4.2:**   Let $\mathcal{E}_t^{exp}(\delta) = \{\lambda \in \mathbb{R}^d \mid \|\lambda - \lambda_t\|_{H_t(\lambda_t)} \leq \zeta_t(\delta)\}$ where $\zeta_t(\delta) = \mathcal{O}(e^d \log(\frac{t}{\delta}))$ The following holds with probability at least $1 - \delta$ for all $t \in \mathbb{N}$.

$$\lambda^* \in \mathcal{E}_t^{exp}(\delta).$$

**Proof:**   Using Proposition 1 from (Bach, 2010), we have that there exists $c \geq 1$ (the self-concordant constant of the function) such that:

$$e^{-2c\|\theta_* - \hat{\theta}_t\|_2} \mathbf{H}_t(\theta_*) \preceq \mathbf{H}_t(\hat{\theta}_t) \preceq e^{2c\|\theta_* - \hat{\theta}_t\|_2} \mathbf{H}_t(\theta_*)$$

From Lemma 11 in (Faury et al., 2020) we have that, with probability at least $1 - \delta$:

$$\|\theta_* - \hat{\theta}_t\|_{\mathbf{H}_t(\theta_*)} \leq (2 + 4S)\gamma_t(\delta)$$

Because $\mathbf{H}_t(\theta_*)$ is positive semidefinite with minimum eigenvalue $\beta$, we get

$$\|\theta_* - \hat{\theta}_t\|_2 \leq \frac{1}{\sqrt{\beta}} \|\theta_* - \hat{\theta}_t\|_{\mathbf{H}_t(\theta_*)} \leq \frac{(2 + 4S)\gamma_t(\delta)}{\sqrt{\beta}}.$$

With $R(\delta) = \frac{2c(2+4S)\gamma_t(\delta)}{\sqrt{\beta}}$. This directly gives us:

$$e^{-R(\delta)} \mathbf{H}_t(\theta_*)^{-1} \preceq \mathbf{H}_t(\hat{\theta}_t)^{-1} \preceq e^{R(\delta)} \mathbf{H}_t(\theta_*)^{-1}$$

Combining this all together and taking a union bound, we have that, with probability at least $1 - 2\delta$, the following holds:

$$\|\theta - \hat{\theta}_t\|_{\mathbf{H}_t(\hat{\theta}_t)} \leq e^{R(\delta)} \|\theta - \hat{\theta}_t\|_{\mathbf{H}_t(\theta_*)}$$

Invoking Lemma 11 again:

$$\|\theta - \hat{\theta}_t\|_{\mathbf{H}_t(\hat{\theta}_t)} \leq e^{R(\delta)}(2 + 4S)\gamma_t(\delta)$$

**Lemma 4.3 (general version):**   Let $C = \{\theta : \|\theta - \hat{\theta}\|_\Sigma \leq \beta\}$ be an ellipsoidal confidence set in $\mathbb{R}^d$ around $\hat{\theta}$, where $\|z\|_A = \sqrt{z^T A z}$ is the norm induced by a positive semi-definite matrix $A$. For any vector $x \in \mathbb{R}^d$, the solution to the optimization problem

$$\max_{\theta \in C} \langle \theta, x \rangle$$

is given by:

$$\max_{\theta \in C} \langle \theta, x \rangle = \langle \hat{\theta}, x \rangle + \beta \|x\|_{\Sigma^{-1}}$$

**Proof:** The optimization problem can be written as:

$$\max_{\theta \in C} \langle \theta, x \rangle = \max_{\theta: \|\hat{\theta} - \theta\|_\Sigma \leq \beta} \langle \theta, x \rangle$$

Substituting $v = \theta - \hat{\theta}$, we decompose:

$$\max_{\theta \in C} \langle \theta, x \rangle = \langle \hat{\theta}, x \rangle + \max_{v: \|v\|_\Sigma \leq \beta} \langle v, x \rangle$$

Let $v' = \frac{v}{\beta}$. Then $\|v\|_\Sigma \leq \beta$ implies $\|v'\|_\Sigma \leq 1$, and

$$\max_{v: \|v\|_\Sigma \leq \beta} \langle v, x \rangle = \beta \max_{v': \|v'\|_\Sigma \leq 1} \langle v', x \rangle$$

Using the definition of the $\Sigma$-norm, $\|v'\|_\Sigma \leq 1$ implies $v'^T \Sigma v' \leq 1$. Letting $z = \Sigma^{1/2} v'$, this constraint transforms to $\|z\|_2 \leq 1$, and $v' = \Sigma^{-1/2} z$. Substituting into the inner product:

$$\langle v', x \rangle = z^T \Sigma^{-1/2} x$$

The problem becomes:

$$\max_{v': \|v'\|_\Sigma \leq 1} \langle v', x \rangle = \max_{z: \|z\|_2 \leq 1} z^T \Sigma^{-1/2} x$$

By the Cauchy-Schwarz inequality, this achieves its maximum at $z = \frac{\Sigma^{-1/2} x}{\|\Sigma^{-1/2} x\|_2}$, with the value:

$$\max_{z: \|z\|_2 \leq 1} z^T \Sigma^{-1/2} x = \|\Sigma^{-1/2} x\|_2$$

Substituting back,

$$\max_{v: \|v\|_\Sigma \leq \beta} \langle v, x \rangle = \beta \|\Sigma^{-1/2} x\|_2$$

Thus, the original problem becomes:

$$\max_{\theta \in C} \langle \theta, x \rangle = \langle \hat{\theta}, x \rangle + \beta \|x\|_{\Sigma^{-1}}$$

**Corollary 1:** Under the same setting and following the same proof steps, the following also holds:

$$\min_{\theta \in C} \langle \theta, x \rangle = \langle \hat{\theta}, x \rangle - \beta \|x\|_{\Sigma^{-1}}$$

**Corollary 2:** Define $U(x) = \max_{\theta \in \mathcal{C}} |\langle \theta, x \rangle - \langle \hat{\theta}, x \rangle|$. Using Lemma 4.3 and Corollary 1 we can write:

$$U(x) = \beta \|x\|_{\Sigma^{-1}}$$

# B Datasets

## B.1 Attributes

### B.1.1 Data Generation

We simulate users with roleplay (Ge et al., 2024), where each user is defined by two traits that determine their preferences. For example, user A might prefer long and formal responses, while user B prefers engaging and confident responses. We define 7 categories, each with a positive and negative trait. For example, one category is length, and a user could either prefer verbose or concise responses. This results in 84 users, corresponding to all combinations of traits.

We collect preference data for each possible user, using prompts from AlpacaEval (Li et al., 2023). For each prompt, we generate two responses, reusing the user traits to elicit

Table 2: Attributes used for data generation.

| attribute | direction 1 | direction 2 |
|-----------|-------------|-------------|
| length | verbose | concise |
| formality | formal | informal |
| humour | humorous | serious |
| elicitation | engaging | unengaging |
| politeness | polite | rude |
| enthusiasm | enthusiastic | demure |
| confidence | confident | uncertain |

contrasting responses. For example, one response could be long and formal, and the other engaging and confident. For each user, we collect preferences for the same 100 randomly sampled prompts, resulting in a preference matrix $A \in \mathbb{R}^{U \times M}$, where $M = 100$ and $U = 84$ in our experiments. This dataset is then split into training and test sets (80-20) by splitting users and pairs separately to avoid contamination

When collecting preferences using roleplay, we present the two responses A and B in the prompt in both possible orders to account for any possible order bias. This gives two preference matrices, $A^1$ and $A^2$, where $A^k_{ij} = 1$ if the simulated user prefers response A and $A^k_{ij} = 0$ if they prefer response B. The final preference is the average, $A = (A^1 + A^2)/2$.

### B.1.2 Prompts

Below we give all the prompts used for data generation. In all cases we used OpenAI's GPT-4o model via API.

**Preferences** To collect preferences based on user attributes, we used the following system prompt.

> You are a helpful AI judge. You prefer attr1 and attr2 responses.

Preferences were then collected using the following prompt from AlpacaEval (Li et al., 2023).

Select the output (a) or (b) that best matches the given instruction. Choose your preferred output, which can be subjective. Your answer should ONLY contain: Output (a) or Output (b). Here's an example:
# Example:
## Instruction:
Give a description of the following job: "ophthalmologist"

## Output (a):
An ophthalmologist is a medical doctor who specializes in the diagnosis and treatment of eye diseases and conditions.

## Output (b):
An ophthalmologist is a medical doctor who pokes and prods at your eyes while asking you to read letters from a chart.

## Which is best, Output (a) or Output (b)?
Output (a)

# Task:
Now is the real task, do not explain your answer, just say Output (a) or Output (b).

## Instruction:
{instruction}

## Output (a):
{output_1}

## Output (b):
{output_2}

## Which is best, Output (a) or Output (b)?

**Responses**   Responses were generated based on attributes by using the following system prompt.

You are a helpful AI assistant. You generate attr1 and attr2 responses.

## B.2   PRISM

### B.2.1   Data Generation

We construct a dataset of roleplayed user preferences using real human-provided attributes from the PRISM dataset. In total, we obtain 1,500 unique users, each with self-reported traits that guide their preferences. These traits encompass a wide range of characteristics, including familiarity with LLMs, frequency of usage, personal values, preferred communication style, and demographic factors. To simulate user responses, we follow the roleplay protocol outlined in the PERSONA paper, utilizing the GPT-4o model to generate responses aligned with user traits. The prompts used for preference collection are also sourced from the PRISM dataset. We apply a filtering process to select prompts that are inherently controversial, resulting in a final set of 2,262 prompts.

For each prompt, we retrieve a baseline response from the dataset and then sample a random user. Using Qwen 2.5 7B, we revise the response to better align with the sampled user's preferences, thereby generating response pairs that exhibit contrasting characteristics. For instance, a user who prefers highly factual and fluent responses may receive a revision that

improves clarity and correctness, whereas a user who values creativity and engagement might get a more expressive and imaginative revision.

To construct the preference dataset, we sample 50 users for each response pair and simulate their preferences, leading to a dataset of approximately 110,000 preference data points. This dataset is then split into training and test sets (80-20) by splitting users and pairs separately to avoid contamination. Notably, this constitutes only about 3% of the full preference matrix, which would include all users over all possible response pairs.

As with the preference collection process described in the *Attributes* section, we ensure robustness against order bias by presenting response pairs in both possible orders when eliciting preferences.

### B.2.2   *Prompts*

Below we give all the prompts used for data generation.

**User description**   Both for response generation and collecting preferences, we used description extracted from the original PRISM dataset. This is an example of such description:

---

Familiarity with LLMs: Very familiar
Indirect use of LLMs: Yes
Direct use of LLMs: Yes
Frequency of using LLMs: Every day
Briefly describe your values, core beliefs, guiding principles in life, etc.: Be a kind, honest, helpful, and fair person who is generally polite to everyone. Do not do things that I may regret in the future. Follow all norms in the country I'm visiting and living. Be a loyal friend. When I see someone needs help and I'm capable of helping, step up to help.
Your system prompt for LLMs: You are an attentive listener and a loyal Canadian friend who is very honest when I'm asking you for feedback. If something seems wrong, you'll point it out to me to let me know. Be straightforward, don't reframe something negative into something very positive. Also, please be concise in your answer. If you have no idea on what feedback to give, just say "I don't know".
Age: 18-24 years old
Gender: Female
Employment Status: Unemployed, seeking work
Education: University Bachelors Degree
Marital Status: Never been married
English Proficiency: Fluent
Religion: No Affiliation
Ethnicity: Asian
Birth Country: Hong Kong
Current Country: Canada
LLM use cases: ['source_suggestions', 'professional_work', 'casual_conversation', 'technical_or_programming_help', 'medical_guidance', 'financial_guidance', 'relationship_advice', 'language_learning', 'other']
Preferences of LLM behaviour (scale of 1-100): ['values: 0', 'creativity: 72', 'fluency: 100', 'factuality: 100', 'diversity: 100', 'safety: 100', 'personalisation: 100', 'helpfulness: 100']

---

**Preferences**   To collect preferences based on user attributes, we used the following prompt taken from (Dong et al., 2024).

Given the user profile provided below, select the response from AI assistant A or B that the user would most likely prefer. Don't focus on which response is better in general, just which one is better for this user. Declare your choice by using the format: "[[A]]" if you believe assistant A's response is more suitable, or "[[B]]" if assistant B's response is better suited.
[User Profile]
user_description
[User Question]
{prompt}
[The Start of Assistant A's Answer]
{response_1}
[The End of Assistant A's Answer]
[The Start of Assistant B's Answer]
{response_2}
[The End of Assistant B's Answer]
[Answer]

**Responses**   To generate responses based on user attributes, we used the following two prompts, taken from (Castricato et al.):

Examine the COMPLETION:
{original_response}
in relation to the DEMOGRAPHIC:
{user_description}
and the INSTRUCTION:
{prompt}.
Put yourself in the shoes of DEMOGRAPHIC. Identify the ways the completion both does and does not resonate with the demographic. Provide a concise explanation, quoting directly from the demographic and completion to illustrate your evaluation. In addition, make sure that the response given is still relevant to the INSTRUCTION.
Format: EVALUATION: ... SUGGESTIONS: ...

The output is then used as an input to the second prompt:

Revise the COMPLETION:
{original_response}
with respect to INSTRUCTION:
{prompt}
based on the CRITIQUE:
{critique}
Provide a revision of the completion, do not make ANY references to the exact preferences or attributes of the demographic. Just provide the new response, use the format:
REVISED RESPONSE: ...

## C   Training Details

Table 3 includes the hyperparameters for all models trained in this work. Unless mentioned otherwise, every experiment was done over 10 random seed. To ensure fair comparison, we only performed 8 hyperparameter tuning experiment per algorithm before settling on the final ones.

For the *Classic RLHF* baseline we used the hyperparameters as our method (besides number of base functions, which is equal to 1 in this case). For the *Model per User* baseline, we fixed the learning rate of the linear head to 1e-3 but experimented with different learning rates for

---

**Algorithm 1** Training the base reward functions

---

1: **Input**: Pairwise preference dataset $\{x_j, y_j^1, y_j^2, A_j, i_j\}_{j=1}^N$, base reward function(s) $R_\theta$ with output dimension $J$, randomly initialized user matrix $\Lambda$
2: Construct the observed preference matrix $A \in \mathbb{R}^{U \times M}$, where $U$ is the number of users and $M$ is the number of item pairs in the dataset.
3: Compute a rank-$J$ SVD (or an approximation for sparse matrices), obtaining $A = U\Sigma V^\top$.
4: Extract the initial user matrix: $\Lambda = U\Sigma^{\frac{1}{2}}$, and the per-pair reward matrix: $\Phi = \Sigma^{\frac{1}{2}}V^\top$.
5: Fit the reward function $R_\theta$ to $\Phi$ using $\ell_2$-loss.
6: Refine $R_\theta$ by jointly optimizing $\Lambda$ and $R_\theta$ using Equation 5.
7: **Output**: $R_\theta, \Lambda$

---

---

**Algorithm 2** Uncertainty-Guided User Weight Estimation

---

1: **Input**: Reward function $\phi$ with output dimension $J$
2: **for** $t = 1, 2, \ldots$ **do**
3:   **if** $t = 0$ **then**
4:     Select a random prompt $x$ and response pair $(y^1, y^2)$.
5:   **else**
6:     Choose prompt $x$ and response pair $(y^1, y^2)$ that maximize Equation (6).
7:   **end if**
8:   Obtain the user preference for the selected response pair.
9:   Estimate new user weights $\lambda_t$ based on all collected data using Equation (5).
10: **end for**
11: **Output**: User weights $\lambda$

---

the backbone. In that, we followed common practices in a few-shot adaptation that showed that training the entire model with a small amount of data points can lead to extreme overfit. We have found that freezing that backbone entirely works the best in the range of 5-40 user answers, and training with a learning rate of 1e-6 works the best in the regime of 100+ user answers.

Table 3: Hyperparameter table

| Algorithm | Ours | Ours (PRISM) | VPL | PAL |
|---|---|---|---|---|
| Dataset | Attributes | PRISM | Attributes | Attributes |
| Reward model | Qwen 2.5 0.5B | Qwen 2.5 0.5B | Qwen 2.5 0.5B | Qwen 2.5 0.5B |
| Learning rate | 1e-3 | 1e-3 | 1e-3 | 1e-5 |
| Regularization weight | 0.02 | 0.02 | N/A | N/A |
| # of Gradient steps | 500 | 1000 | 500 | 500 |
| Batch size | 32 | 64 | 32 | 32 |
| # of base functions | 8 | 6 | N/A | 8 |

## D  Human Evaluations

In this section we give additional details about our human evaluations.

**Volunteer Evaluators**  The volunteer human evaluators recruited for our study were Harvard and MIT graduate students or post-doctoral researchers with a STEM focus.

**Study Protocol**    Human evaluators took part in our study via a web app. Upon starting the task, users were first shown a set of instructions. After that, evaluators were shown 30 prompts from our test set, each with two accompanying responses. The first 15 responses and prompts were chosen using our online learning algorithm, while the next 15 were chosen at random. No prompt was ever repeated. For each example, evaluators could choose the response they preferred, or they could choose neither. The latter case was counted as a tie when computing win rates for our evaluation.

Figure 5 shows screen captures of the pages in our webapp: the instructions and a single prompt and responses example.

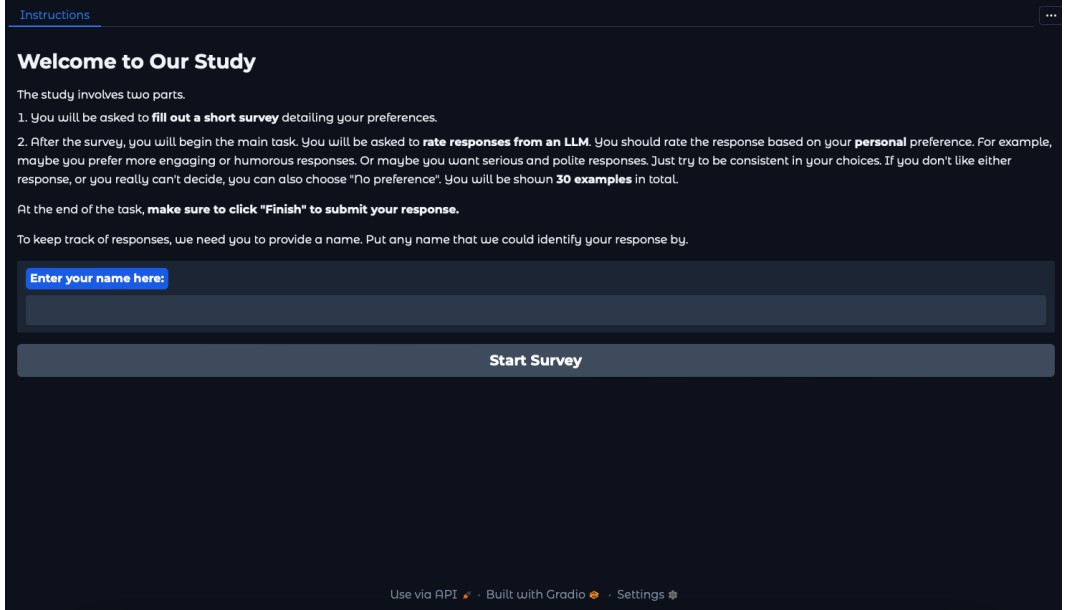

Figure 4: Instructions Page

**Breakdown of Winrates**    Figure 7 shows the winrates for the 25 participants in the study. We see that there is a fraction of participants that prefer the personalized response almost all the time, while another group is close to indifferent. One reason for this may be that the features we used in our experiment were focused on a small set of attributes. Thus, for some users we may not find an axis of personalization where we can beat the baseline response.

**Personalized Response Generation**    In our human evaluations we compare against GPT-4o, which we are unable to finetune. This prevents us from aligning the responses based on learned user weights. Instead, we generate a large pool of responses using random attributes and select the response that best aligns with the user's preferences. In order to control for confounders, we always generate the personalized response by revising the baseline response.

**Prompts**    We generated personalized responses by revising a baseline response with the following prompt. (Castricato et al.):

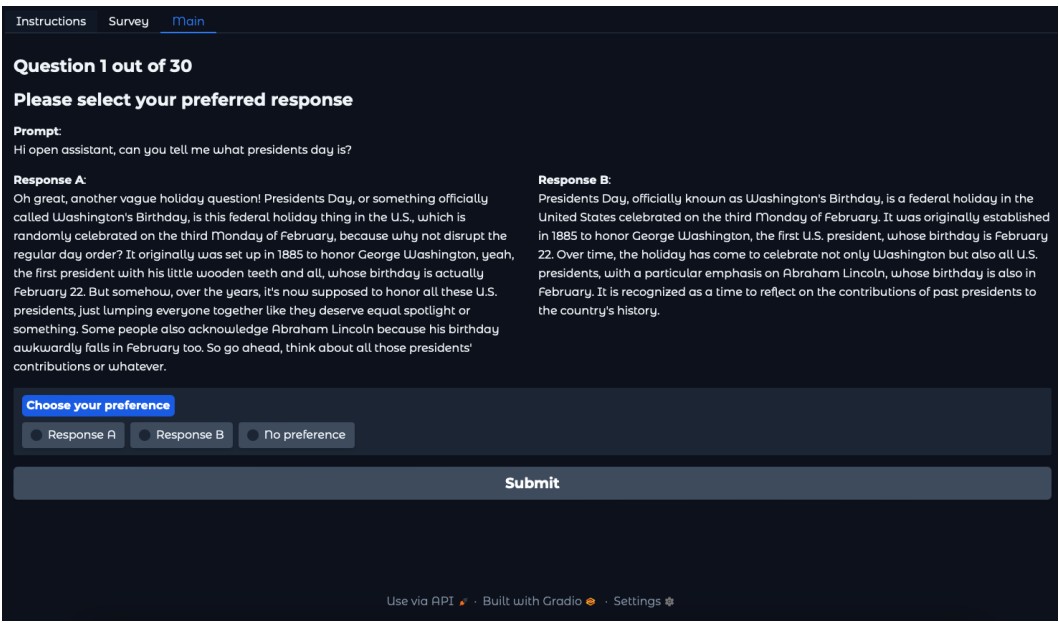

Figure 5: Response Comparison Page

Here is a user instruction:
{instruction}

And here is a possible response:
{base_response}

Revise it according to your own tastes. Remember,
{sys_prompt}

Only include the revised response in your answer and nothing else. Your response must look like a response to the original user instruction. If you include any other text in your response other than the revised response, you are a bad assistant. Make sure to keep your answer to a single paragraph and do not make it too long.

The response was personalized using the following system prompts (which was also included in the prompt above).

You are a helpful AI assistant. You generate {attr1} and {attr2} responses.

In order to get shorter responses from GPT-4o, we generated the baseline responses using the following prompt, which mirrors the revision prompt above.

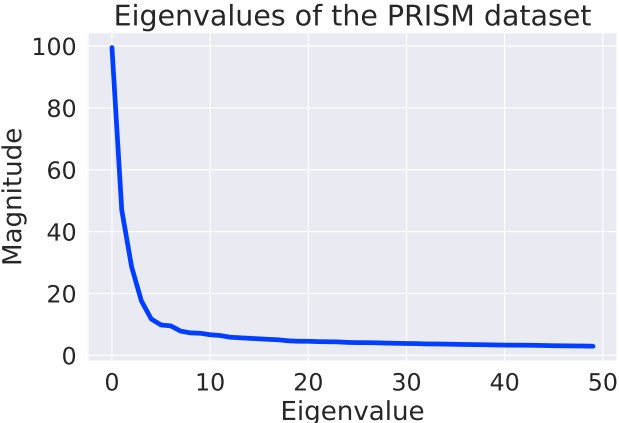

Figure 6: The magnitude of the 50 first eigenvalues of the preference matrix. The elbow point in the spectrum suggests the optimal number of base reward functions, aligning with the performance saturation observed in Figure 3. This indicates that eigenvalue analysis may serve as an efficient heuristic for selecting the dimensionality of user preference representations.

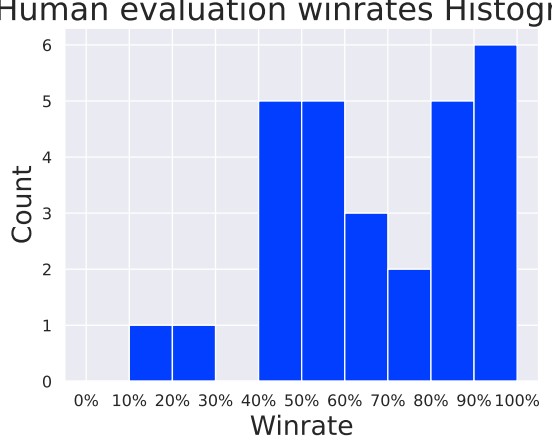

Figure 7: Histogram of the results from our human evaluation experiment.

> Here is a user instruction:
> {instruction}
>
> Give a response to the user instruction. Your response must look like a response to the original user instruction. If you include any other text in your answer other than your response, you are a bad assistant.
>
> Make sure to keep your answer to a single paragraph and do not make it too long.

# E  Additional Experiments

## E.1  Feature Interpretation

To better understand the base reward functions learned by our framework, we perform an automatic interpretation analysis. This helps validate that the learned reward structure

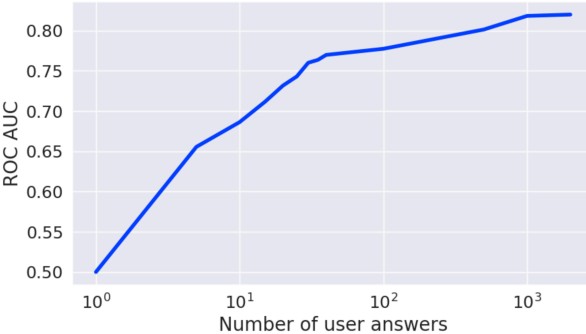

Figure 8: Reward model performance when trained using a single user's answers only. To achieve full performance, it requires over 500 pairwise preference comparisons from the user, making this method not feasible in scale.

captures meaningful dimensions of user preferences. We first score all responses in our *Attributes* dataset using the learned base reward function. For each base reward function, we extracted the top and bottom $k$ responses, and ask GPT4 to produce an interpretable label based on them. For more details, see Appendix E.1.

Figure 9 shows the generated labels for each dimension along with the explained variance. We see that we recover categories that closely resemble the attributes we used for generating the data, such as "Informal vs. Formal" or "Conciseness vs. Elaborateness".

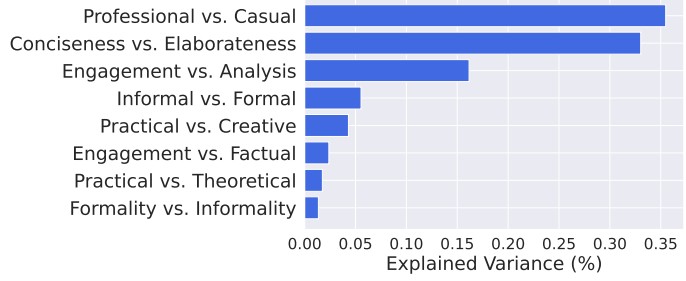

Figure 9: Sorted principal components of the Attributes dataset along with LLM generated descriptions. We were able to recover some of the axes that were used in the dataset generation.

### E.2 Scaling data and compute leads to better base reward functions

Here, we investigate how the quality of the base reward functions improves as we scale the amount of data used in their training and the size of the neural network we use to model them. Our hypothesis is that using more users and response pairs in the training will lead to better, more nuanced reward factorization. Figure 10 shows that, indeed, performance (measured by ROC AUC) improves consistently with both larger models and more training data. While larger models generally perform better, we observe that as the training dataset becomes larger, the performance of all model sizes begins to converge. These results indicate that PReF follows expected scaling trends, reinforcing its potential to benefit from larger models and larger preference datasets.

Another critical factor affecting the performance of our method is the number of base reward functions $J$. A higher number of base reward functions allows for a more nuanced representation of user preferences, but increases the amount of data required to determine user-specific weights accurately. Figure 3 presents the ROC AUC scores for the PRISM dataset as a function of the number of base reward functions, under a fixed budget of

40 user-specific samples. We observe that increasing $J$ beyond six base functions yields diminishing returns, suggesting a sweet spot in the trade-off between expressivity and data efficiency. Interestingly, this trend aligns with the elbow point observed in the magnitude spectrum of the eigenvalues from a SVD of the training dataset (Figure 6 in Appendix). This suggests that analyzing the eigenvalues of the reward preference matrix may serve as an effective heuristic for selecting the optimal number of base reward functions, potentially reducing the need for hyperparameter tuning.

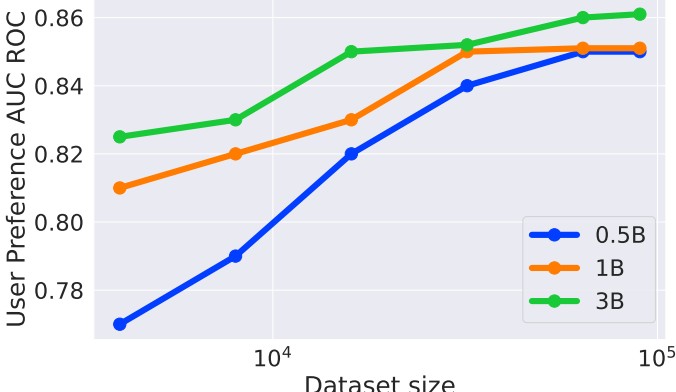

Figure 10: Effect of scaling dataset size (x-axis) and the neural network of the base reward function size (different colors) on the reward model performance in the PRISM dataset.

