# OpenReview forum: "Language Model Personalization via Reward Factorization"
_colmweb.org/COLM/2025/Conference — COLM 2025_

### Official Review · Reviewer_Fo71 · 2025-05-11

**Rating:** 5
**Confidence:** 3
**Ethics Flag:** 1

**Summary:**

The paper proposes an elegant way to personalize Reward Models per user preferences. To "onboard" new user, few (<=30) prompt-response pairs are necessary per user and the method uses uncertainty-based selection strategy to determine a pair should be presented to the user.

The authors claim 67% win rate improvement over GPT-4o. It's not very clear from the paper but my understanding is that 67% win rate is GPT-4o vs GPT-4o + inference-time alignment method with Qwen-based personalized RM. I'd appreciate if the authors make it more obvious.

**Questions To Authors:**

Lines 261 - 262. How exactly to you split the data into (train, validation, calibration, test) set. For example, in the test set, are there unique users not present in other splits? Or each user can have some prompt-response pairs in more than one split?

**Reasons To Accept:**

Interesting approach to tackle personalization.

**Reasons To Reject:**

Main downside is that the proposed approach is for Reward models. Which makes onboarding new users possible only if LLM is then aligned using inference-time alignment technique.

It would be interesting to see how this approach compares to a much simpler baseline: just give GPT-4o these 10-30 collected preference pairs *in-context* and see if it will be able to tailor it responses per user better.

---

> ### Author Response · Authors · 2025-06-01
>
> > It would be interesting to see how this approach compares to a much simpler baseline: just give GPT-4o these 10-30 collected preference pairs *in-context* and see if it will be able to tailor it responses per user better.
>
> We completely agree that this is an important baseline to consider. In fact, we included this exact experiment in our paper using Qwen2.5-7B on the PRISM dataset (see lines 319–321). To further strengthen this point, we’ve now added results using GPT-4o on the Attributes dataset. Specifically, we provide GPT-4o with the user’s preference pairs as in-context examples and compare it to our method (PReF). The win rates of PReF over the in-context learning baseline are as follows:
>
> We agree this is a valuable baseline. In fact, we included this exact comparison in the paper using Qwen2.5-7B on the PRISM dataset (see lines 319–321). To further strengthen this point, we now include additional experiments using GPT-4o on the Attributes dataset. Specifically, we provide GPT-4o with the user’s preference pairs as in-context examples and compare its outputs to those generated via GPT4o+PReF. The win rates of PReF over in-context learning are:
>
> - 5 responses: **49.8%**
> - 10 responses: **55.3%**
> - 20 responses: **59.4%**
>
> These results suggest that while in-context personalization performs reasonably well with a handful of examples, it plateaus quickly. In contrast, PReF continues to improve as more data is collected, indicating its stronger generalization and personalization capabilities.
>
> > Main downside is that the proposed approach is for Reward models. Which makes onboarding new users possible only if LLM is then aligned using inference-time alignment technique.
>
> We agree with the reviewer that our approach assumes access to a reward-conditioned alignment method at inference time - this is indeed a requirement and a constraint. However, we believe this is a reasonable tradeoff, especially when compared to the alternatives for personalization.
>
> In-context learning, while convenient, quickly saturates and fails to fully capture nuanced preferences (see above). Fully fine-tuning a model per user is prohibitively expensive, both computationally and in terms of data requirements (see Figure 8).
>
> In contrast, PReF strikes a balance:
>
> - It requires only a small number of user interactions.
> - It amortizes all training costs across users.
> - It enables *dynamic* personalization at inference time.
>
> While inference-time alignment techniques do add complexity, they are increasingly common [1,2,3,4], and we view this dependency as a practical design choice for enabling scalable and data-efficient personalization.
>
> > How exactly to you split the data into (train, validation, calibration, test) set.
>
> We thank the reviewer for highlighting this important point. As explained in lines 261–264, our data splits are structured as follows:
>
> - **Training Set:** Contains a subset of users and prompt-response pairs.
> - **Validation Set:** Contains responses to new prompts for the same users present in the training set.
> - **Calibration Set:** Introduces entirely new users (not present in training or validation sets) responding to prompts already seen during training. This set is specifically used to calibrate the model for these new users.
> - **Test Set:** Includes the same users from the calibration set but presents them with entirely new prompts, ensuring no overlap with prompts from the calibration set.
>
>
> [1] Mudgal, Sidharth, et al. "Controlled decoding from language models."
>
> [2] Han, Seungwook, et al. "Value augmented sampling for language model alignment and personalization."
>
> [3] Rame, Alexandre, et al. "Rewarded soups: towards pareto-optimal alignment by interpolating weights fine-tuned on diverse rewards."
>
> [4] Khanov, Maxim, Jirayu Burapacheep, and Yixuan Li. "Args: Alignment as reward-guided search."

---

> > ### Comment · Reviewer_Fo71 · 2025-06-08
> >
> > thank you for you comments. I'll keep the score.

---

> > > ### Author Response · Authors · 2025-06-08
> > >
> > > Dear Reviewer,
> > >
> > > Thank you for taking the time to read our rebuttal and for your thoughtful comments. Since we included the specific experiment you suggested and addressed the concerns you raised, we would be grateful to understand if there's anything else you feel is missing or could be improved to make the work publishable in your eyes.
> > >
> > > Best regards,
> > > The authors

---

### Official Review · Reviewer_wGH9 · 2025-05-12

**Rating:** 7
**Confidence:** 5
**Ethics Flag:** 1

**Summary:**

The authors introduce a new method called "Personalization via Reward Factorization (PReF)" that factorizes each individual user's reward as a linear combination of base functions. While prior works required training user-specific reward models, the authors demonstrate that PReF is more data efficient and achieves higher win rate across two datasets.

**Questions To Authors:**

In Section 5.3, I am curious about the reasons why PReF performs better than VPL and PAL. Beyond the quantitative results, could the authors provide thoughts on why user-provided system message and ICL isn't effective?

**Reasons To Accept:**

1. Section 3 and Section 4 provide a good explanation of what problem the authors are addressing and and theoretical guarantees of how they designed their PReF method to achieve that.

2. The ablation experiments provide sufficient insights on how to design a good personalization system (L2 regularization, SVD initialization, and how the framework could be extended as more number of user's responses are accumulated).

**Reasons To Reject:**

1. In Table 2 of Appendix B, the authors define 7 distinct attributes. I think the core of LLM personalization papers is a good definition of different user attributes that could well capture what people care about from LLM responses in real-world interactions. Yet, the attributes that the authors test are confined to a narrow set of stylistic-related attributes. There should be more consideration of what users might take into account beyond stylistic features and an inspection of PReF could also capture those. (I am aware that the authors also tested on the PRISM dataset, but am mentioning this to check if they have any thoughts on which attributes the PReF method could be extended to).

2. Related to 1, in a real-world setting, users might not explicitly verbalize what they prefer or the preferences might be entangled with each other and even contradict (e.g., In setting A, user X might prioritize preference M than preference N and in setting B, user X might prioritize preference N than preference M). The settings that the authors employed (Attributres, PRISM) do not seem to capture this and remains to be quite synthetic. I don't think that this is what the paper should solve, but yet, since the method assumes that we could extract user weights to define a user specific reward, it is concerning of whether this method could be extended in these more practical settings.

---

> ### Author Response · Authors · 2025-06-01
>
> > Reasons to reject.
>
> These are excellent points, and they go to the heart of what motivated our work.
>
> We completely agree that real-world personalization goes well beyond stylistic preferences. As we noted in the paper, the Attributes dataset was designed as a controlled toy setting to evaluate PReF in isolation. It allowed us to validate the method’s ability to recover known user preferences in a simplified space. But our broader vision, and what PReF is designed to support, is learning personalization axes directly from user behavior\data, rather than hand-crafting them.
>
> This is why we evaluated on the PRISM dataset as well, which includes richer, messier, and more human preference signals (e.g., alignment with values, tone, subjectivity). We acknowledge that these are still imperfect proxies for fully realistic settings, but they’re a step toward evaluating PReF in more open-ended, high-dimensional preference spaces.
>
> Regarding context-dependent or even contradictory user preferences: this is a great point, and PReF is actually well-equipped to handle it. Because the base reward functions are learned from data and not tied to fixed attributes (like “conciseness” or “humor”), they can encode prompt-conditional behaviors. For example, if a user prefers verbose responses for factual prompts and concise ones for personal prompts, PReF can learn two separate base functions that score those dimensions. The user’s preference vector would assign weight to both, and PReF would resolve which one to activate based on the prompt content.
>
> We do acknowledge that fully capturing this richness depends on having training data with sufficient coverage of such nuanced preferences. Collecting such data remains a valuable direction for future work.
>
> > Questions To Authors.
>
> We appreciate the reviewer raising this insightful question. From our experiments with the PRISM dataset, we observed the following. Regarding **User-Provided System Messages**, we found that users often struggle to articulate precise nuances of their preferences clearly and effectively in natural language. For instance, expressing subtle distinctions ("I prefer short responses, but not too short - more like concise and clear...") is inherently challenging, leading to vague or suboptimal system prompts.
>
> While **ICL** works relatively well when provided with only a small number of examples (as shown in Table 1), it struggles significantly as the number of examples grows. We believe that an advantage of our method is that although it uses the same number of data points during inference, the training on many prompts and users' preferences enables accurate adaptation.

---

> > ### Comment · Reviewer_wGH9 · 2025-06-09
> >
> > Thank you for your response!
> >
> > As mentioned in my initial review, exploring more realistic dimensions of preferences seems most crucial (and the bottleneck) to personalization research but this is clearly out of scope for this paper and the authors have already conducted a lot of experiments on the paper. I believe that the contribution is significant and the responses are very thoughtful! I will raise my score, hoping that this paper could be presented at the main conference.

---

### Official Review · Reviewer_fVG8 · 2025-05-15

**Rating:** 8
**Confidence:** 4
**Ethics Flag:** 1

**Summary:**

This paper introduces a framework called PReF to utilize the low-rank structure of user preferences. PReF addresses the research problems: 1. learning the low-rank reward models with seen users; 2. active learning for efficiently learning new users' preferences. This paper shows PReF can adapt to a new user and predict their reward better than previous methods. It also validates the personalized responses according to the learned reward function with both static datasets and human evaluation.

**Questions To Authors:**

* In Section 4.1, why don't we have an L2 regularization term in Eq. (5) as in Eq. (6). Logistic Matrix Factorization does have L2 regularization. I wonder if we can overcome the instability challenges with regularization. We can still initialize via the corresponding WALS [1] solution.

* Again for learning the base functions, have the authors considered constraining $\lambda \geq 0$ or modeling the reward function as a convex combination of the base functions like LoRe [2]. LoRe (without active learning) also outperforms both VPL and PAL.

* As said, I don't understand how we pick the prompt $x$ with Eq. (9) during active learning and how we find $y^1$ and $y^2$ using the eigenvector.

* In Figure 3(A), did we require to tune the value of $\beta$?

* In Algorithm 2 L6: ``maximize Equation (6)`` should be ``maximize Equation (9)``.

* In Algorithm 2 L9:  ``using Equation (5)`` should be ``using Equation (8)``.

[1] Y. Hu, Y. Koren, and C. Volinsky. Collaborative ltering for implicit feedback datasets. In ICDM 2008, pages 263--272.

[2] Avinandan Bose et. al.. LoRe: Personalizing LLMs via Low-Rank Reward Modeling. https://arxiv.org/abs/2504.14439

**Reasons To Accept:**

* This paper comprehensively studies almost all problems of language model personalization and proposes reasonable solutions.

* The active learning part is novel and has theoretical support.

* Table 1 shows the proposed method has better reward prediction accuracy than VPL and PAL.

* This paper really demonstrates the quality gains of personalized responses. It also uses two alignment methods: Best-of-N and VAS, showing the generalizability of the proposed reward modeling.

* The human evaluation also supports the quality of personalized responses.

**Reasons To Reject:**

* For the adoption or active learning part, I see maximizing Eq. (7) or (9) by enumeration can work but don't see the proposed method being more practical. In particular, I don't understand how we pick the prompt $x$ before using Eq. (9). Given $x$, I don't understand how we pick a response pair using an inference time alignment algorithm. In particular, how do we guarantee that we can exactly solve $\phi(x, y^1) = \frac{1}{2}\nu$. Why don't we use $\phi(x, y^1) = 2\nu$ and $\phi(x, y^2) = \nu$ for example?  Also, by comparing Figure 3(C) and Table 1, it looks like random selection already outperforms both baselines. Typically confidence set based method is conservative like other bandit algorithms but also like to see if there is any approximation in maximizing Eq. (9).

* I feel the SVD initialization is a little bit ad-hoc also wonder if we have a better way to learn the base functions. See my questions below.

---

> ### Author Response · Authors · 2025-06-01
>
> We thank the reviewer for the thoughtful and constructive feedback. We're glad to hear that they found our work comprehensive and promising, and we appreciate the recognition of both the theoretical contributions and practical results.
>
> > For the adoption or active learning part, I see maximizing Eq. (7) or (9) by enumeration can work but don't see the proposed method being more practical...
>
> The problem with maximizing equation (9) by enumeration is the huge space of possible responses which makes this calculation step expensive. Instead, we solve Eq. (9) analytically to obtain the optimal direction *ν*, which is the top eigenvector of the uncertainty-weighted feature space. Once we have *ν*, we use inference-time alignment methods that can generate a response *y₁* such that ϕ(x, y₁) ≈ +0.5ν, and a response *y₂* such that ϕ(x, y₂) ≈ −0.5ν. This gives a feature difference of roughly *ν*, as desired. This makes the approach scalable and feasible in real-world settings, avoiding expensive enumeration.
>
> Why ±0.5ν? This choice is arbitrary in principle. Any pair (a, b) such that ϕ(x, y₁) − ϕ(x, y₂) = ν would suffice, including your suggestion of (2ν, ν). We picked ±0.5ν to center the responses symmetrically around the origin, which empirically helped with generation stability, but we will clarify in the paper that other scalings are equally valid.
>
> > In Section 4.1, why don't we have an L2 regularization term in Eq. (5) as in Eq. (6). Logistic Matrix Factorization does have L2 regularization. I wonder if we can overcome the instability challenges with regularization. We can still initialize via the corresponding WALS [1] solution.
>
> You're absolutely right to highlight this. We agree that L2 regularization is a standard and important component in stabilizing training. To clarify, during the **SVD-based initialization** phase, we do *not* apply L2 regularization. This step is purely used to obtain a structured, low-rank initialization of Λ and Φ from the observed (binary) preference matrix A. Once we move to the **MLE refinement phase**, regularization is *indeed applied* to mitigate issues like scale ambiguity and to prevent degenerate solutions.
>
> This distinction between the two stages (unregularized SVD init and regularized MLE optimization) was not made clear enough in the original text. We will explicitly note this in the camera-ready version.
>
> Finally, we appreciate the suggestion to consider WALS-style initialization. While we did not implement weighted regularization in the SVD phase, incorporating it is a promising future improvement and may further enhance the stability of the initial factorization.
>
> > Again for learning the base functions, have the authors considered constraining λ≥0 or modeling the reward function as a convex combination of the base functions like LoRe [2]. LoRe (without active learning) also outperforms both VPL and PAL.
>
> We appreciate the reviewer pointing us to this recent work. Since it was published only a month after the COLM deadline, we were not able to address it in the paper. We will cite and discuss it in the camera-ready.
>
> > As said, I don't understand how we pick the prompt x with Eq. (9) during active learning and how we find y1 and y2 using the eigenvector.
>
> In our implementation, prompt *x* is sampled from a pool of pre-selected prompts used in the dataset (e.g., from the test or calibration set). In principle, one could select *x* to maximize expected information gain (e.g., by computing the highest-variance eigenvector *ν* across prompts), but we found that simply sampling *x* uniformly and selecting the most informative response pair *given x* worked well in practice.
>
> > In Figure 3(A), did we require to tune the value of β?
>
> We found $\beta=0.02$ to work sufficiently well on all of our experiments in this paper. Further tuning could improve performance, but we haven't try.

---

> > ### Comment · Reviewer_fVG8 · 2025-06-07
> > **Rebuttal acknowledgement**
> >
> > I acknowledge that I have read the authors' response and I am maintaining my score. The proposed framework for modeling preference and user adoption is principled and justified by the results.

---

### Official Review · Reviewer_iLD4 · 2025-05-16

**Rating:** 4
**Confidence:** 3
**Ethics Flag:** 1

**Summary:**

This paper proposes PReF, a framework for LLM personalization via reward factorization. It assumes that user preferences lie in a low-dimensional space and can be expressed as a linear combination of shared base reward functions. The method requires only a small number of user responses for effective adaptation and integrates with existing inference-time alignment techniques without modifying model weights. Experiments on both synthetic and real data show promising results, with a 67% win rate over GPT-4o in human evaluations.

**Questions To Authors:**

1.	How robust is the heuristic for selecting the number of base reward functions, and how does it impact performance?
2.	Can users inspect or adjust their personalization parameters to improve transparency or control?
3.	How does this method align with broader safety and value alignment goals?
4.	How well does the approach generalize beyond synthetic datasets to more diverse, natural user interactions?

**Reasons To Accept:**

1. Tackles the limitations of existing alignment methods by modeling user-specific rewards through shared base components.
2. Needs only around 10 examples per user and avoids model finetuning, making it practical for large-scale use.
3. Demonstrates strong results in both synthetic and real-user settings, with effective use of active learning.

**Reasons To Reject:**

1. May not capture complex or evolving user preferences.
2. It remains unclear how interpretable or complete these learned components are.
3. The framework lacks mechanisms to track and update changing user preferences.

---

> ### Author Response · Authors · 2025-06-01
>
> We would like to respond to the points the reviewer raised:
>
> > Reasons To Reject: (1) May not capture complex or evolving user preferences. (2) It remains unclear how interpretable or complete these learned components are. (3) The framework lacks mechanisms to track and update changing user preferences.
>
> - Complexity of User Preferences - Our framework is designed to flexibly accommodate complex user preferences. Increasing the number of base reward functions allows us to model finer-grained preference dimensions (see Fig. 3B), without overfitting or degrading performance.
>
> - Interpretability - we refer to Appendix E.1, where we show that the learned base reward functions can indeed be interpreted and analyzed qualitatively, even though they are data-driven.
>
> - Changing User Preferences - We agree that tracking evolving user preferences is important. To support evolving preferences, users can be periodically re-queried using our adaptive algorithm. The lightweight nature of the adaptation, requiring only ~10 examples, makes this practical. Alternatively, the system can provide the user with the option to provide feedback while using the LLM.  In general, our paper presents a framework and not a specific system, and is quite flexible in the way that it can be used.
>
> > How robust is the heuristic for selecting the number of base reward functions, and how does it impact performance?
>
> We refer the reviewer to Figure 3(B), where we checked how the number of base reward functions impacts performance. Choosing a higher number of base functions typically does not harm performance but only increases computational cost. We see the main goal of the heuristic (Figure 6 in the appendix) as a way to get a lower bound on the number of reward functions. Therefore, the heuristic is robust because it provides a conservative guideline rather than a strict constraint.
>
> > Can users inspect or adjust their personalization parameters to improve transparency or control?
>
> Absolutely. While we focus on automatic adaptation in our experiments, the personalization vector λ can be exposed to users through interpretable UI elements (e.g., sliders for reward dimensions), enabling interactive control over response style or alignment priorities. This again highlights the flexibility of our framework, which can support different systems.
>
> > How does this method align with broader safety and value alignment goals?
>
> Our method is complementary to established safety and value alignment techniques. By conducting experiments primarily with already-aligned models, we demonstrate that our personalization effectively serves as a second alignment step, after safety RLHF. An interesting future direction is to analyze the differences between this 2 step approach to a combined RLHF training of both safety and personalization.
>
> > How well does the approach generalize beyond synthetic datasets to more diverse, natural user interactions?
>
> While we did not conduct experiments with fully natural user interactions, we deliberately designed our experimental setup to closely approximate such settings:
>
> - The prompt datasets used in both the PRISM and Attributes evaluations were sourced from actual user interactions with LLMs, ensuring that the inputs reflect natural usage patterns.
> - The model responses were generated directly from the aligned base models, exactly as they would be in a deployed system.
> - Although user preferences were synthetically defined, we introduced realism through injecting noise into the preference labels by using sampling (rather than greedy decoding).
>
> These choices were made to ensure that our synthetic setup closely mirrors real-world deployment scenarios. The strong results we observe under these conditions give us confidence that our method will generalize well to natural settings.

---

> > ### Comment · Reviewer_iLD4 · 2025-06-11
> >
> > Thank you for your response. I believe interpretability is central to the contribution of your work, so referring readers to Appendix E.1 is not ideal. While most of my comments have been addressed or partially touched on, some key points remain insufficiently justified. For example, the claim that “the lightweight nature of the adaptation, requiring only ~10 examples, makes this practical” lacks empirical support. Therefore, I will maintain my current rating.

---

### Decision · Program_Chairs · 2025-07-08

**Decision:**

Accept

**Comment:**

The authors propose a new way of personalizing the output of LLMs: by factoring user preferences in a linear, low dimensional space of reward functions, and combining them on a per-user basis. Experiments demonstrate strong sample efficiency compared to baselines on 2 datasets.

Reviewers raised some concerns that were partially addressed in response.

iLD4's biggest remaining concern is that appendix E1 about interpretablility is not in the main text body: perhaps the authors could surface a pointer in revision?

Fo71 suggested an additional experiment, which was only partly covered in the original submission. The authors added the additional experiment which further shows the strengths of their approach.

Overall, I am pleased with the discussion between reviewers and authors, and believe many of the biggest concerns have been addressed. This work represents an interesting contribution above and beyond existing works, and (while some presentation factors could be improved) many of the shortcomings have been addressed in discussion.